# Remote monitoring of seismic swarms and the August 2016 seismic crisis of Brava, Cape Verde, using array methods

Carola Leva[1], Georg Rümpker[1] and Ingo Wölbern[1]

[1]Institute of Geosciences, Goethe-University Frankfurt, Altenhöferallee 1, 60438 Frankfurt am Main, Germany

*Correspondence to*: Carola Leva (leva@geophysik.uni-frankfurt.de)

**Abstract.** During the first two days of August 2016 a seismic crisis occurred on Brava, Cape Verde, which – according to observations based on a local seismic network – was characterized by more than thousand volcano–seismic signals. Brava is considered an active volcanic island, although it has not experienced any historic eruptions. Seismicity significantly exceeded the usual level during the crisis. We report on results based on data from a temporary seismic–array deployment on the
neighbouring island of Fogo at a distance of about 35 km. The array was in operation from October 2015 to December 2016 and recorded a total of 1343 earthquakes in the region of Fogo and Brava, 355 thereof were localized. On 1 and 2 August we observed 54 earthquakes, 25 of which could be located beneath Brava. We further evaluate the observations with regards to possible precursors to the crisis and its continuation. Our analysis shows a significant variation in seismicity around Brava, but no distinct precursory pattern. However, the observations suggest that similar earthquake swarms commonly occur close
to Brava. The results further confirm the advantages of seismic arrays as tools for the remote monitoring of regions with limited station coverage or access.

## 1 Introduction

The islands of the Cape Verde archipelago are located about 700 km west of the coast of Senegal on top of the Cape Verde Rise, which originates from a mantle plume (Courtney and White, 1986). Brava is the westernmost island of the southern
chain, see Fig. 1. Although considered active, no volcanic eruptions occurred on Brava since the settlement in the 15th century. On its main plateau, pyroclastic deposits and phreatomagmatic craters are associated with recent volcanic activity on Brava, probably of Holocene age (Madeira et al., 2010). The characteristics of phreatomagmatic activity pose a potential threat to the ~6000 inhabitants of the island. Volcanic unrest is documented in degassing studies and in the high seismicity beneath and around the island. High degassing of deep–seated $CO_2$, mainly in the northeast, has been linked to magmatic processes (Dionis
et al., 2015). The seismicity is dominated by volcano–tectonic earthquakes and shifts over time in location and frequency, as can be seen from past studies, e.g. Helffrich et al. (2006), Faria and Fonseca (2014) and Vales et al. (2014). In contrast to Brava, the neighbouring island Fogo shows only minor seismic activity. Fogo is located about 20 km east of Brava. Fogo volcano is erupting at mean intervals of about 20 years. The last eruption took place from November 2014 to February 2015 (González et al., 2015; Cappello et al., 2016; Richter et al., 2016; Calvari et al. 2018).


On 1 and 2 August 2016, a seismic crisis occurred beneath Brava. According to data recorded by a permanent seismic monitoring network, the crisis comprised about 1000 shallow earthquakes and tremors (Faria and Day, 2017). Authorities decided to evacuate about 300 inhabitants from two villages (ECHO, 2016). As access to the aforementioned data is restricted by government (Faria, personal communication), we used recordings from a simultaneously operating seismic array on Fogo to analyze the earthquake activity during the crisis and extended the analysis to the period between October 2015 and December 2016. In this study we report to which extend a seismic array can be used for remote monitoring of a volcanic seismic crisis and present the seismicity beneath and around Brava. To gain information about possible precursors of this crisis and about the further development of the seismicity after the crisis, the observation of the shift of the earthquake locations in the months before and after the crisis will be emphasized.

## 2 Seismic network and data

From October 2015 to December 2016 we operated a seismic array on Fogo which served as a pilot study in preparation for a larger multi–array installation that started in 2017. The pilot array of 2016 consisted of 10 seismic stations, arranged in two circles around a central station with an aperture of 700 m. Two stations were vandalized and one failed; the remaining seven stations were equipped with short–period 4.5 Hz geophones, see Fig. 2b. The array was designed for an analysis of local events with mean frequencies of 7.5 Hz, the array transfer function for the reduced array is shown in the supplementary Fig. S1. In order to allow for classical detection and localization techniques, we deployed three additional broadband sensors on Fogo island, see Fig. 1. These stations were only used to better locate events beneath Fogo. All stations are equipped with CUBE–data loggers and powered with 12 V batteries. Data were recorded continuously at a sample rate of 200 Hz. Some data gaps occurred due to the limited storage capacity of the data loggers (as indicated in Fig. 3 below).

## 3 Methods

### 3.1 Array analysis

Array techniques provide a suitable tool to locate events at distance outside of the network and can also be applied to events without clear onset of phases. The latter is the case for typical seismic signals associated with volcanic activity, such as tremors, long period or hybrid events (e.g. Wassermann, 2012). While the network is located on Fogo, most earthquakes occur beneath and around Brava, mainly at distances of about 35 km from the array. The position of the seismic stations relative to the earthquakes leads to a large uncertainty when applying classical localization procedures, thus the earthquakes around and beneath Brava are located using the array. The purpose of array techniques is to improve the signal–to–noise ratio (SNR) by beamforming, i.e. by time shifting and stacking the coherent part of the signal (e.g. Schweitzer et al., 2012). The beamforming can be applied in the frequency– or in the time–domain. Here, we perform the beamforming in the time–domain, which is computationally more expensive, but incorporates a broad frequency band. Also, by first time–shifting the phases, we are able to select a rather narrow time window around the phases of interest which improves the coherency of the stack, even if the

(initial) separation between the onsets of common signals at different stations is relatively large (see Singh and Rümpker (2020) for details).

Array analysis is based on the assumption that the event is located sufficiently far away from the array so the incoming
wavefront can be treated as a plane wave (Schweitzer et al., 2012) traversing the array with a specific backazimuth and apparent velocity. Beamforming is utilized to determine the horizontal slowness components ($s_x$, $s_y$), which also yields the backazimuth of the event. From the inverse of the absolute slowness, the apparent velocity ($v_a = 1/|s|$) is determined. To obtain the horizontal slowness components, a grid search is applied. We consider a range between -0.3 s/km and 0.3 s/km for both $s_x$ and $s_y$ with a grid size of 64×64. The slowness limits correspond to reasonable values for expected apparent velocities of incoming
wavefronts from local events. The array traces are shifted according to all possible slowness values defined by the grid and are summed up in the time–domain. From the maximum of the total energy (given by the integrated squared amplitudes) of the sum trace, we obtain the slowness and backazimuth of the first arriving P–wave.

The initial data processing involves application of a Butterworth filter to improve the SNR of the recordings. The cutoff–frequencies are chosen from a spectral analysis of the event and are applied to all traces. We perform a time–domain array
analysis by choosing a time window of about ten times the dominant period (i.e. one or two seconds in most cases) around the onset of the P–wave (see Fig. 2a). Then the traces are time shifted according to the given slowness components. The next step is to define a shorter stacking window (within the first window) that spans one or two periods of the P–wave arrival and is used to calculate the total energy of the sum trace. Ideally, the total energy reaches a maximum if the time–shift of the P–wave arrival across the array is properly accounted for by the given slowness. In Fig. 2a and Fig. 2c this stacking window is marked
in red. Both time windows are selected in reference to the central array station. The trace of the central station itself is kept fixed, while the remaining traces are shifted with respected to the given slowness and distance from the central stations. The energy as a function of slowness components is displayed in Fig. 2d. From the slowness components ($s_x$, $s_y$) that correspond to maximum energy, the absolute slowness and the backazimuth are determined by $s = \sqrt{s_x^2 + s_y^2}$ and $BAZ = 90° - \arctan(s_x/s_y)$, respectively. To estimate the error of the backazimuth we choose a 95% level around the maximum peak of
energy. The maximum and minimum backazimuth within this energy level are then selected as errors, typically leading to uncertainties of about 10° in the backazimuth of the earthquakes in our study.

### 3.2 Epicentral–distance estimates

The array localization does not provide information about the epicentral distance and the event depth. Here, the distance between an event and a station is determined from the S–P travel–time difference. First, the theoretical arrival times of P– and
S–waves are estimated by using a two–layer model with mean velocities of 6.1 km/s and 8.0 km/s representing the crust and the upper mantle, respectively (in view of Vales et al., 2014). We assume the Moho at a depth of 14 km and a fixed event depth of 5 km, in line with previous studies of events near Brava (Faria and Fonseca, 2014). Even though some authors (Vales et al., 2014) reported a larger event depth of about 10 km, the error which results from the uncertainty of the depth is well within the

error of the distance estimation (see below). From the P– and S–wave travel–time curves for this model, differential arrival times are obtained as a function of epicentral distance. During the localization process the epicentral distance is determined for each array station. From the distance values the mean distance and standard deviation is computed. The error is estimated by evaluating the effect of the different parameters on the distance calculation for the two–layer model. For this purpose, we systematically varied the crustal and mantle velocities, the event depth and the Moho depth. It turns out that the variation of the crustal velocity has the largest impact on the distance estimation. For the events of interest, we generally assume a minimum distance error of 10% which exceeds the error due to variations of the crustal velocity. This error thus incorporates the uncertainties of the simple two–layer assumption, including the uncertainties due to event– and Moho depth and the velocities. Only, if the standard deviation of the distance (as described above) is even larger, we assign this as the error. However, this applies only to a few events. Typically, the absolute error in epicentral distance is in the range of 5 to 8 km.

Note, that the distance estimation used here is appropriate for events that occur within the crust. To ensure that this is the case, the apparent velocity derived from the array analysis is used as an indicator, as it corresponds to the velocity at the ray turning point. An apparent velocity within the range of typical crustal velocities thus indicates a ray path that is confined to the crust (see Leva et al., 2019).

## 4 Results

During the study period our stations on Fogo recorded mainly volcano–tectonic earthquakes that occurred beneath and around Brava. We were able to analyze a total of 355 earthquakes. The volcano–tectonic earthquakes exhibit frequencies typically between 10 Hz to 30 Hz (Fig. 4). The magnitudes generally range between 0.7 and 2.7, however, the smallest magnitude is 0.3 and the highest 3.7. On average we recorded four earthquakes per day (Fig. 3). The precise locations, magnitudes, and errors of each analyzed event are given in the supplemental material along with maps that contain the error ellipses. The seismicity is characterized by highly variable earthquake locations over the period of more than one year. To better constrain the variation of the seismicity close to Brava in the time before the seismic crisis in the beginning of August 2016, we analyzed the locations of earthquake clusters month by month. In the following we will describe the changes in seismicity over time and emphasize the occurrence of earthquake clusters and periods with elevated seismicity (Fig. 3).

### 4.1 October 2015 to July 2016 – before the seismic crisis

In October 2015 we observed a peak in seismicity (see Fig. 3) with two earthquake clusters (Fig. 5a). One cluster occurred on 8 and 9 October and was located southwest of Brava. From 10 to 15 October the second cluster to the northwest of Brava became active. On 19 October the dominant seismic activity occurs again in the area southwest of Brava, where it remained until 23 October. After that, we observed a shifting back to the position of the second cluster northwest of Brava.

On 12 November, the number of recorded earthquakes per day reached 13 (Fig. 3) exceeding the average number of earthquakes per day, but the locations of the earthquakes were rather widespread in the north of Brava (see Fig. 5a). In February the earthquakes shifted to an area west of Brava with an increased seismicity on 18 and 19 February (Fig. 5b). From 7 to 11

April a high seismic activity was recorded with events originating from an area extending from southwest offshore Brava about 20 km towards south–southeast (Fig. 5b). Seismicity reached another peak on 10 May (Fig. 3), but the locations remained in the area south of Brava until August (Fig. 5b). From April to June the southern station of the array was out of operation, leading to a possible bias in earthquake locations. A more detailed analysis shows that true locations are somewhat closer to Brava (by about 8 km) than shown here. Seismic events during June still are located mainly offshore south of Brava (Fig. 6). A data gap occurred from 17 June to 18 July due to limited storage capacities of the data loggers. During the last days of July, we observed very few earthquakes distributed over a wider area (Fig. 5b).

### 4.2 August 2016 – during the seismic crisis

On 1 and 2 August the seismic crisis occurred on Brava (Faria and Day, 2017). We detected 54 earthquakes during these two days and were able to locate 25 individual events of this swarm. Most of the volcano–tectonic earthquakes are located beneath the southern part of Brava (Fig. 7a). The magnitudes ranged from 0.5 to 2.8 and the b–value is 0.83 (Fig. 8c). We estimated the b–value following Gutenberg and Richter (1944) with $\log_{10} N = a - bM$, where $N$ represents the cumulative number of earthquakes with magnitudes larger than $M$. The magnitude of completeness is determined using the maximum curvature method (Wiemer and Wyss 2000), as this method has been shown to be relatively reliable for catalogues with small sample sizes (Mignan and Woessner, 2012). The constants $a$ and $b$ are obtained from fitting the Gutenberg–Richter relation for values above the magnitude of completeness. However, the b–value is difficult to estimate with certainty, as the number of earthquakes is relatively low (see Roberts et al., 2015). This is underlined by the variation of $N$ (blue) with respect to the straight line (black) fitted to the data (Fig. 8). The analysis of a possible temporal evolution of the b–value is added to the supplementary material.

In the aftermath of the crisis, seismicity remained at an elevated level. Until 15 August, earthquakes were again located west and south offshore, but relatively close to the island (Fig. 7b). Afterwards the seismicity around Brava decreased and was distributed over a broader area. On 15 August, a swarm of deep subcrustal earthquakes occurred beneath Fogo. Due to their proximity to the array, the earthquakes were analyzed and located by conventional network–based methods only (using the additional network stations on Fogo). These deep events are further discussed in Leva et al. (2019).

### 4.3 September 2016 to December 2016 – after the seismic crisis

In September and October, we still observed earthquakes beneath Brava (see Fig. 9a), but they did not cluster locally and the seismic activity was relatively low during this time. An elevated level of seismicity was recorded on 12 and 13 November, extending from west to south offshore of Brava (Fig. 9b). From 29 November to 2 December we recorded a total of 150 earthquakes (see Fig. 3). On 29 and 30 November, a swarm was located directly northwest of the coast of Brava (Fig. 9b). In the following two days the seismic activity shifted towards the south to an area south–west of Brava's coast. During the rest of December, earthquakes mainly occur beneath the southern part of Brava and offshore the southern coast (Fig. 9b).

## 4.4 Periods with increased seismicity

Figure 3 indicates, that periods with elevated seismicity frequently occur beneath and around Brava during the time of our study. Apart from the swarms in August 2016, we observe four additional peaks, where the records show more than 20 earthquakes per day: 9 to 15 October 2015, 7 to 11 April, 10 May and 29 November to 2 December 2016. These earthquakes have in common, that they occur offshore (Fig. 10). B–values for the earthquakes during 9 to 15 October 2015 and 29 November to 2 December 2016 are 1.28 and 0.9, respectively (Fig. 8b,d). For the other two periods the number of earthquakes was too low to determine the b–value.

## 5 Discussion

The seismicity beneath and around Brava is characterized by a significant variation in the location of the highest activity. This also becomes evident when comparing the results of previous studies, which show different areas of high seismic activity around Brava (e.g. Heleno and Fonseca, 1999; Helffrich et al., 2006; Faria and Fonseca, 2014; Vales et al., 2014). In our study from October 2015 to December 2016 we observe several periods with increased seismicity (Fig. 3), which originate from different areas. During the first months of 2016 we observe a shift of the volcano–tectonic earthquakes from west of Brava (during February to March) towards an area south of Brava (during April to July). On 1 and 2 August a seismic crisis occurred on Brava. According to Faria and Day (2017) about 1000 shallow earthquakes and tremors were recorded by the local seismic network on Brava. We observed 54 earthquakes with our network on Fogo and were able to locate 25 earthquakes with magnitudes from 0.5 to 2.8. The discrepancy in the number of detected earthquakes is due to the distance of about 35 km between our network and the area of high seismic activity. Small earthquakes are therefore masked by seismic background noise. Also, in our data, we do not observe tremors or long–period events, originating on Brava. However, we cannot exclude the occurrence of such events, as they may not be detectable at the distance of the array. However, from the magnitude–frequency relation (see Fig. 8c) we can estimate that magnitudes must have been as low as -1 to reach the high number of events detected by Faria and Day (2017). For our observations, the magnitude of completeness is 1 and the b–value 0.83. Due to the small number of earthquakes in the swarm, it is possible that the b–value is underestimated (as also discussed further below). The locations of earthquakes that we observed cluster mainly southwest of Brava at a distance of about 3.5 to 4 km south of the reportedly evacuated village Cova de Joana. However, considering the errors in our localization (see Fig. 7b), the main activity may indeed have occurred close to the village as indicated by the results of Faria and Day (2017). The array analysis is not suited to observe a possible depth migration of the events. In the aftermath of the crisis most earthquakes still arise beneath Brava, in October the dominant seismic activity shifts back to the regions west and south of Brava, where it remains until December 2016.

Seismic arrays can exhibit systematic aberrations, which may influence the localization of seismic events. In order to determine a possible systematic deviation from the true earthquake locations, we compare the backazimuth and slowness values of the array analysis with those obtained by classical network analysis at a later time (e.g. Schweitzer et al., 2012). Within a more

comprehensive study from January 2017 to January 2018, we operated a seismic network consisting of three arrays and seven
single stations equipped with short–period sensors on both Fogo and Brava (see Fig. S3). The shape and location of the array
AF in that study coincides with the array used during the pilot study presented here. By determining the systematic aberration
of array AF, we can therefore draw conclusions for the location accuracy of both arrays. For earthquake locations on Brava,
we determine a mean deviation of the backazimuth of about 6.5° towards the south. Further details of the analysis are given in
the supplemental material. Figure 11 shows the resulting new locations of the earthquakes during the seismic crisis, after taking
the correction into account. As a result, the earthquake locations tend to be shifted closer to the village of Cova de Joana.

Our observation of a shift in earthquake locations from west to south of Brava prior to the crisis does not provide evidence for
a distinct precursory signal related to the seismic crisis in the beginning of August, especially when considering the days just
before the crisis, for which we observe seismicity distributed over a broad area. Another point is the variation in seismic
activity afterwards, especially from November to December, which again shows a shift from the west to the south without
invoking another crisis. During the time of our experiment, a dispersed occurrence of earthquake clusters seems to be rather
common in the study area. Faria and Day (2017) report on a change in seismicity around Brava after an earthquake of
magnitude M4 in September 2015. However, as their data are restricted, we cannot comment on this observation in detail.

As depicted in Sect. 4.4, during the time of our study we record four additional periods, apart from the swarms in August,
where the number of earthquakes exceeds 20 per day. These times with elevated seismicity occur from 9 to 15 October 2015,
7 to 11 April, 10 May and 29 November to 2 December 2016. From 9 to 15 October 2015, the dominant seismic activity occurs
northwest of Brava. Clustering of earthquakes only occurs during the period from 29 November to 2 December. For these two
periods the b–values are estimated as 1.28 and 0.9, respectively. However, this estimation is rather uncertain, as the number
of earthquakes is low (see Roberts et al., 2015) and the detections performed by the array are biased towards larger events.
The magnitudes of completeness are difficult to assess and the corresponding b–values are likely underestimated, even when
considering the whole study period for which we estimated a b–value of 0.8 (Fig. 8a). High b–values significantly above 1
would be expected for volcano–tectonic earthquake swarms (Roberts et al., 2015). To better constrain the underlying processes,
analyses of focal mechanisms are helpful, but not available due to limited azimuthal coverage provided by the array. The
observed clustering and frequent variations in earthquake locations are characteristic of volcano–tectonic earthquake swarms
(e.g. Zobin, 2012) and their origin is likely attributed to magmatic processes, as also suggested by other authors (e.g. Faria and
215 Fonseca, 2014). In previous studies of volcano–tectonic earthquakes their origin, often, is attributed to dyke inflation or dyke
propagation (Roman and Cashman, 2006). Earthquake swarms without a typical mainshock–aftershock sequence usually occur
in response to fluid migration or volatile and $CO_2$ releases, causing reduced fault resistance or stress changes (e.g. Lindenfeld
et al., 2012). In previous studies examining the seismicity of Cape Verde, earthquake swarms west of Brava have been linked
to a shallow volcano–tectonic structure with a NE–SW alignment between the Cadamosto Seamount and Brava (Vales et al.,
2014). Earthquake swarms NE and SW offshore Brava have been associated to submarine volcanic cones and earthquakes
close to or beneath Brava to magmatic intrusions into the crust (Faria and Fonseca, 2014).

Comparing the additional periods of elevated seismicity with the seismic crisis in the beginning of August, it seems that the potential risk for the population on Brava may have been increased during the seismic crisis, as earthquake locations cluster beneath the island. However, we cannot determine the depth of the events, which is another crucial parameter in estimating the potential hazard. The occurrence of this seismic crisis on a dormant volcano characterized by previous phreatomagmatic activity clearly underlines the importance of the local monitoring network, which has been established in 2011 (Faria and Fonseca, 2014). There are several documented cases of failed eruptions accompanied by swarms of volcano–tectonic earthquakes at dormant volcanoes. A failed eruption is characterized by magma intrusion into the upper crust, accompanied e.g. by seismic swarms, which stops without an eruption (Moran et al., 2011). These volcanic unrests are indistinguishable from unrests leading to eruptions, which makes a forecast difficult (Zobin, 2012). For example, in 1989 an unrest of Mammoth Mountain, California, was documented on the basis of increased seismic activity with several earthquakes clusters active at different episodes with rather small magnitudes ($M \leq 3$), long–period and very–long–period earthquakes, together with outgassing of magmatic $CO_2$ and fumaroles with increased $^3He/^4He$–ratios. This unrest has been interpreted as ascent of magma from the mid–crust to the upper crust (Hill and Prejean, 2005). Therefore, a possible scenario for the mechanisms leading to the seismic crisis on Brava could be that magma has been transported into the upper crust, where the process came to a halt. Diffuse carbon dioxide ($CO_2$) degassing surveys have been regularly conducted on Brava during the period from 2010 to 2018 (Albertos et al., 2019), and the observed spatial–temporal changes on ground $CO_2$ efflux value and diffuse $CO_2$ emission rates are geochemical evidences which support a volcanogenic source for the 2016 anomalous seismic activity registered at Brava (García-Merino et al., 2017; Albertos et al., 2019).

Taken together our observations of 2016 and the observation of a change in seismicity after a large earthquake in September 2015 (Faria and Day, 2017), the seismicity following in 2016 could potentially be part of an uplift episode. As reported by Madeira et al. (2010) and Ramalho et al. (2010a), Brava experienced significant uplift, which cannot be explained by a regional uplift across the Cape Verde swell. Magmatic intrusions below the volcanic edifice could cause this uplift (Ramalho et al., 2010a,b). A failed eruption could contribute to such an uplift, however we cannot comment on the amount of material added and thus on a potential uplift.

The village of Cova de Joana on Brava is in the vicinity of a volcano-tectonic lineament and it has been suggested that the volcanism on Brava could be controlled by tectonic stresses (Madeira et al., 2010). Also an interaction of regional tectonic and volcanic stresses, as observed at El Hierro, Canary Islands (López et al., 2017) could be a possible mechanism causing the earthquakes beneath Brava. The clear identification of the mechanism behind the events during the seismic crisis and their relationship to faults on Brava would require more precise locations, and focal mechanisms of the earthquakes, in addition to observations from other disciplines such as geochemistry and geodesy.

## 6 Conclusions

In this study we remotely monitored a seismic crisis by tracking the shifting of swarms of volcano–seismic events using array methods. We observe changes in seismic activity before, during and after the seismic crisis. In general, seismic arrays are valuable tools for the remote seismic monitoring of regions that are difficult to access.

The array of this study was located on Fogo, Cape Verde, about 35 km apart from the neighboring island of Brava, and was operational from October 2015 to December 2016. We analyzed the seismic crisis that occurred on Aug. 1 and 2 on Brava and observed an elevated level of seismicity. 54 earthquakes were detectable on those two days, 25 could be located. During the first six months of 2016 the seismicity around Brava shifted over time from a region located offshore west of Brava to another offshore area south of the island. During this time, the number of earthquakes per day exceeded 20 earthquakes per day during three periods (9 to 15 October 2015, 7 to 11 April and 10 May 2016). However, during these periods the earthquakes occur offshore and in a rather large area. In the last days of July we recorded only very few earthquakes, which we located in a widespread area around and beneath Brava. This leads to the conclusion, that we did not find any evidence for seismic precursors of the crisis, such as a shift of the volcano–tectonic earthquakes towards the island.

After the two days of the seismic crisis the activity beneath Brava remained at an elevated level until October, where we find a widely distributed seismicity around and beneath Brava. During the end of November and the beginning of December another swarm of earthquakes occurred offshore west of Brava. Thus, it appears that the seismicity shifted away from the island again. We conclude that the seismic crisis might be an example of a failed eruption, likely caused by the transport of magma and/or $CO_2$ into the upper crust, as it has been suggested by the observed changes on diffuse $CO_2$ degassing surveys (García-Merino et al., 2017; Albertos et al., 2019).

Although the seismic array used in this study provided important independent information about the seismic crisis on Brava in August 2016, the inclusion of additional (e.g. geochemical and geodetical) data is highly desirable and required. In general, the combination of different observables could significantly improve the assessment of volcanic hazards.

### Data availability

The data is available for download at GEOFON (https://geofon.gfz-potsdam.de). Please refer to Wölbern et al., 2019.

### Author contributions

CL analyzed the data and prepared the figures for the here presented work. CL wrote the manuscript as part of her PhD under supervision of GR. The manuscript was reviewed and edited by GR and IW. The study and the setup of the seismic array were initiated and conceived by GR and IW. IW was also responsible for project administration. All authors took part in the field work.

**Competing interests**

The authors declare that they have no conflict of interest.

**Acknowledgements**

We thank Bruno Faria for his support and we would like to acknowledge the efforts of José Levy in customs handling and
logistics. Paulo Fernandes Teixeira and José Antonio Fernandes Dias Fonseca are thanked for their assistance during field
work. The Geophysical Instrument Pool Potsdam provided three seismic instruments of the pilot study in 2016 and the
instruments of the main study in 2017. We further thank Carmen López and an anonymous reviewer for their comments and
suggestions, which helped to improve the manuscript.

**Financial Support**

The project was funded by Goethe University Frankfurt and by Deutsche Forschungsgemeinschaft (DFG) through grants to G
Rümpker and I. Wölbern (WO 1723/3-1), respectively.

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

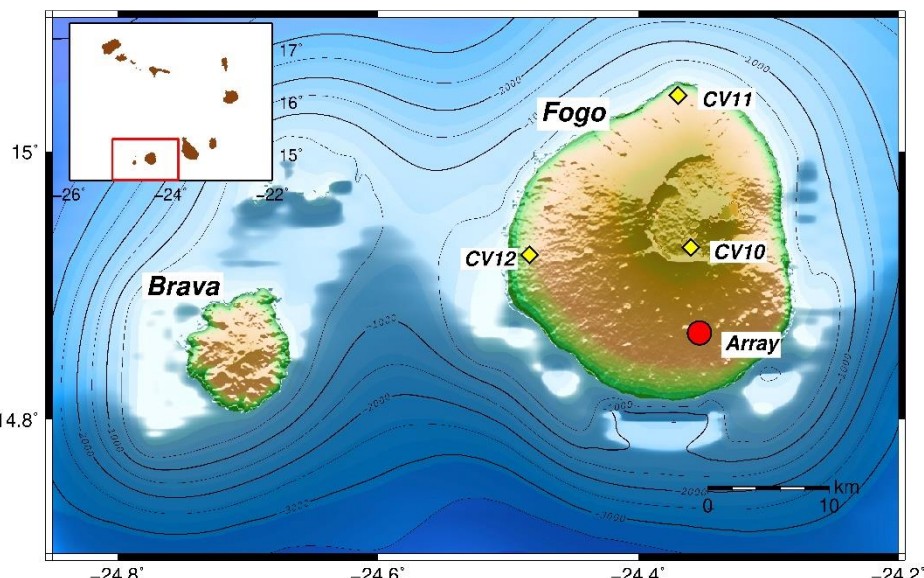

**Figure 1: Location of the island and station locations on Fogo. Circle: location of the array, which consists of ten stations, of which seven were operational. The array was operated from October 2015 to December 2016. Diamonds: additional single broadband stations. These were operational from January 2016 to December 2016. Inset top left: map of Cape Verde, red rectangle: current map section of Brava and Fogo. Topographic and bathymetry data are from Ryan et al. (2009).**


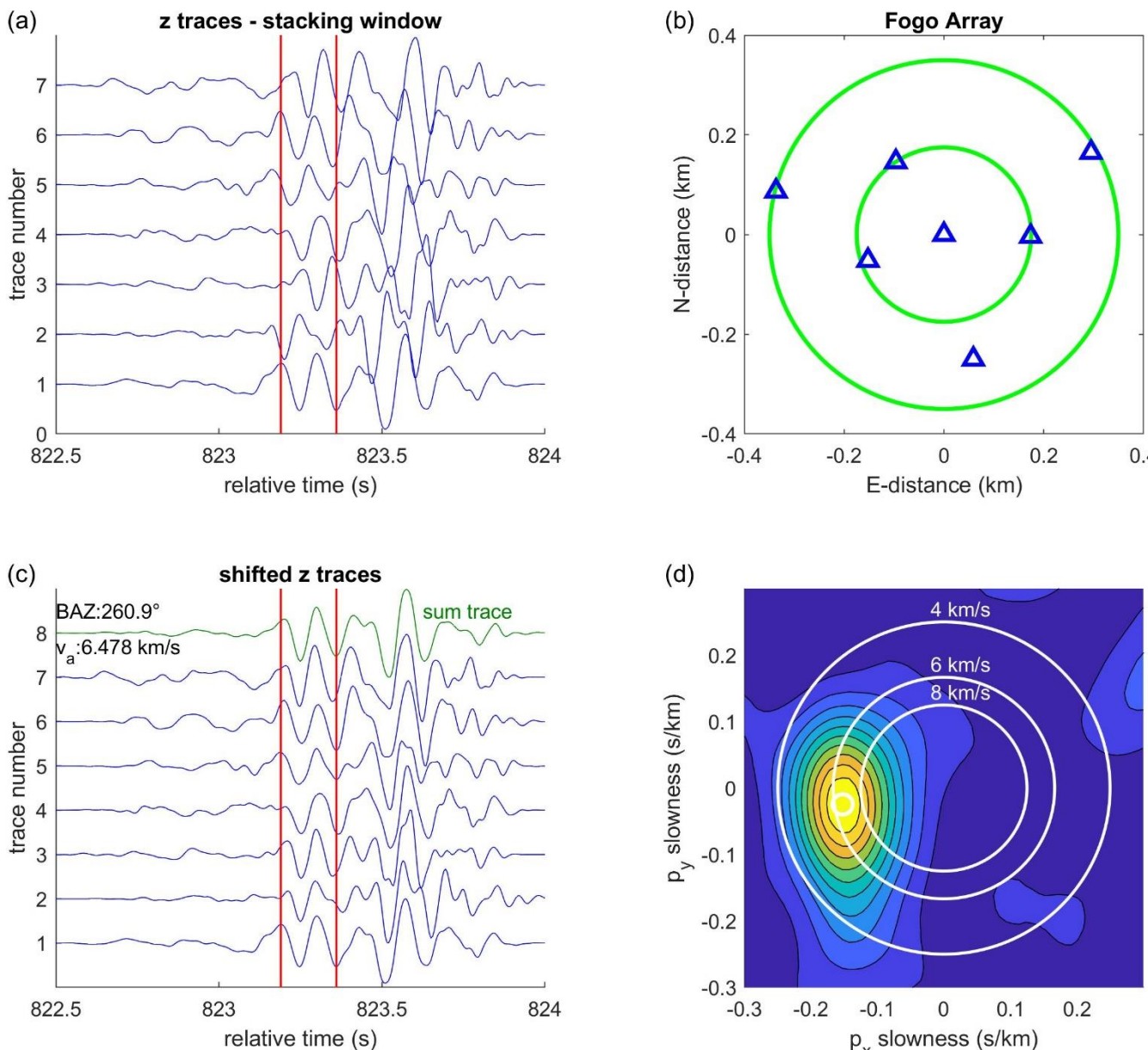

**Figure 2: Example of the array analysis applied to an event of the seismic crisis (2 August 2016, 01:13 (UTC)). (a) Record section before shifting and stacking. Traces of the seven array stations are filtered between 1 and 18 Hz. The time window has a length of 2.5 seconds, the smaller stacking window is marked in red. (b) The configuration of the seismic array. (c) Traces after shifting and stacking. The sum trace is marked in green. (d) Time–domain energy stack.**


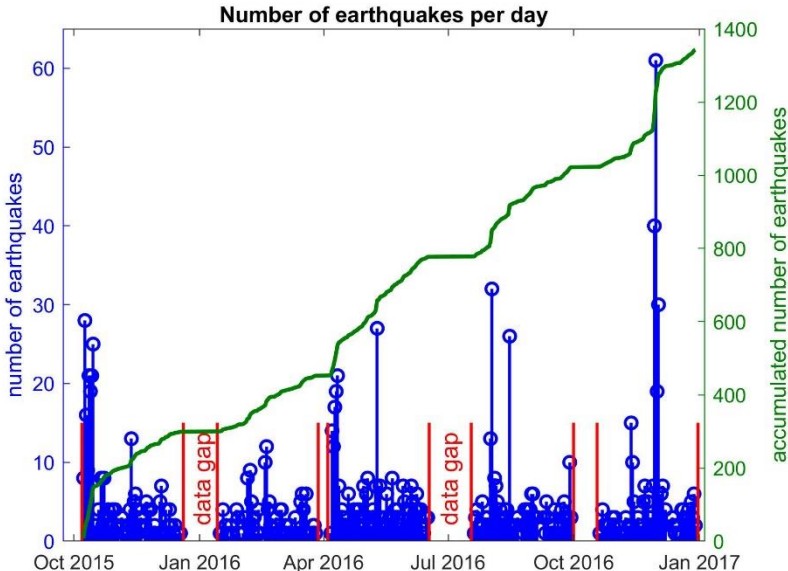

**Figure 3: Blue:** Number of detected earthquakes per day from October 2015 to December 2016. **Green:** accumulated number of earthquakes. Red lines indicate periods with data gaps.

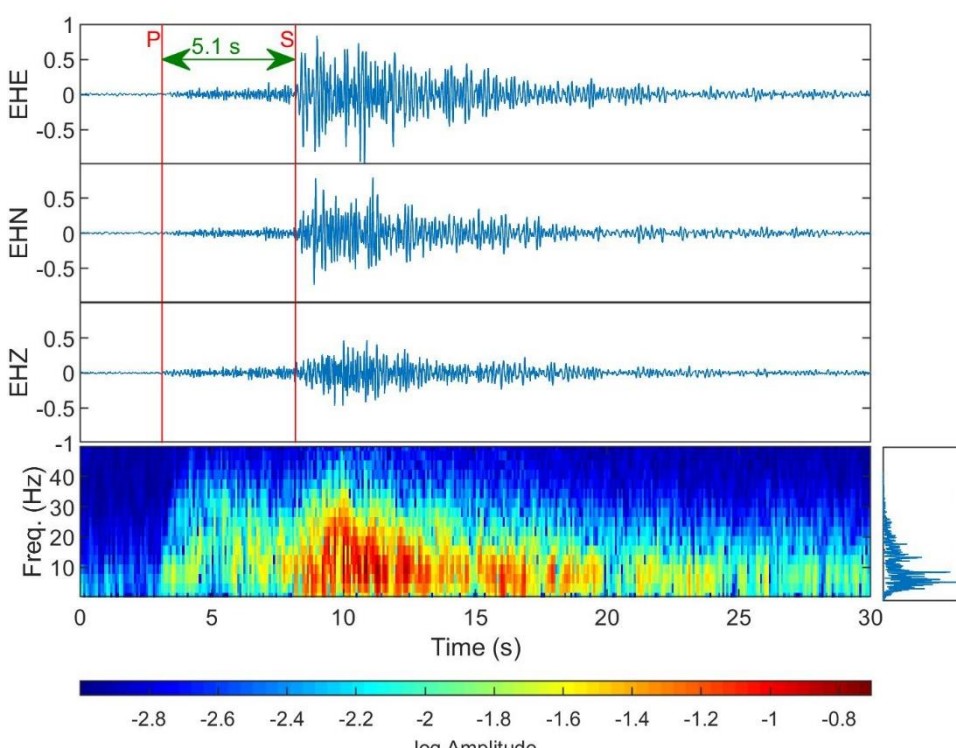

**Figure 4: Top: example of a typical earthquake near Brava, recorded on 2 August 2016, 01:13 (UTC) at a short–period station of the array on Fogo. A Butterworth filter is applied with cutoff–frequencies of 0.5 to 50 Hz and traces are normalized. Bottom left: spectrogram of the vertical component. Bottom right: frequency content of the recording.**

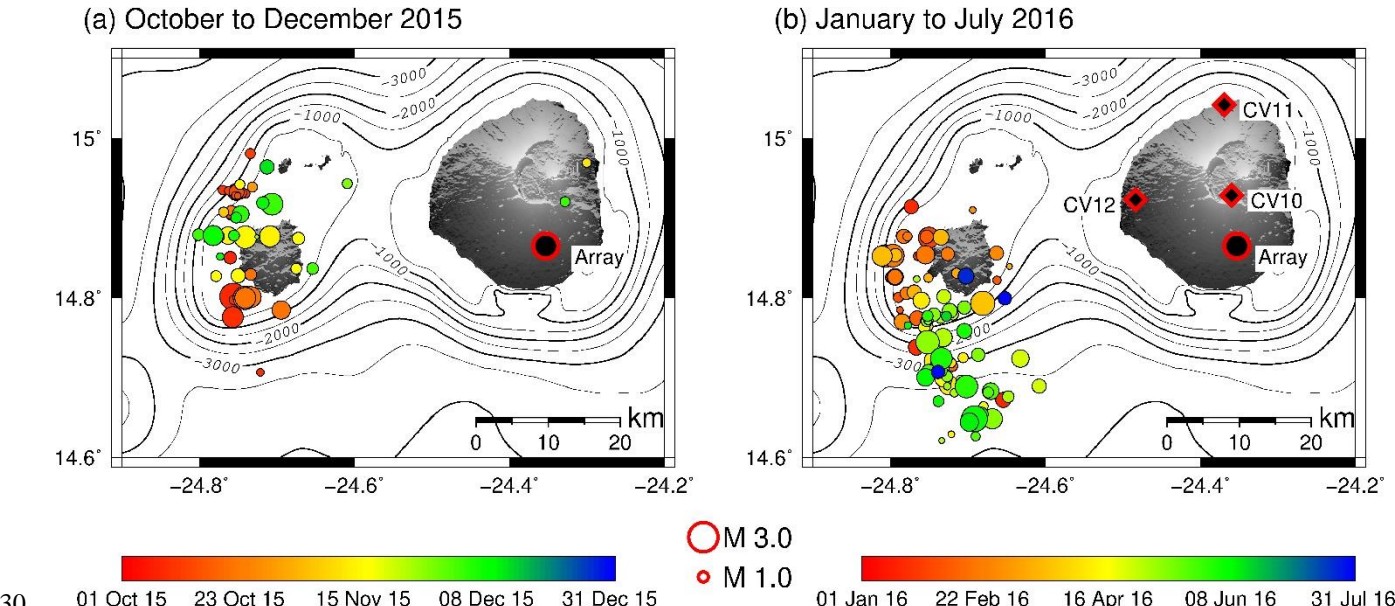


**Figure 5: (a) Earthquake locations from 8 October to 19 December 2015. Red/black circle: position of the array on Fogo. (b) Earthquake locations from 15 January to 31 July 2016. Red/black circle: position of the array, red/black diamond: additional broadband stations on Fogo. Topographic and bathymetry data are from Ryan et al. (2009).**

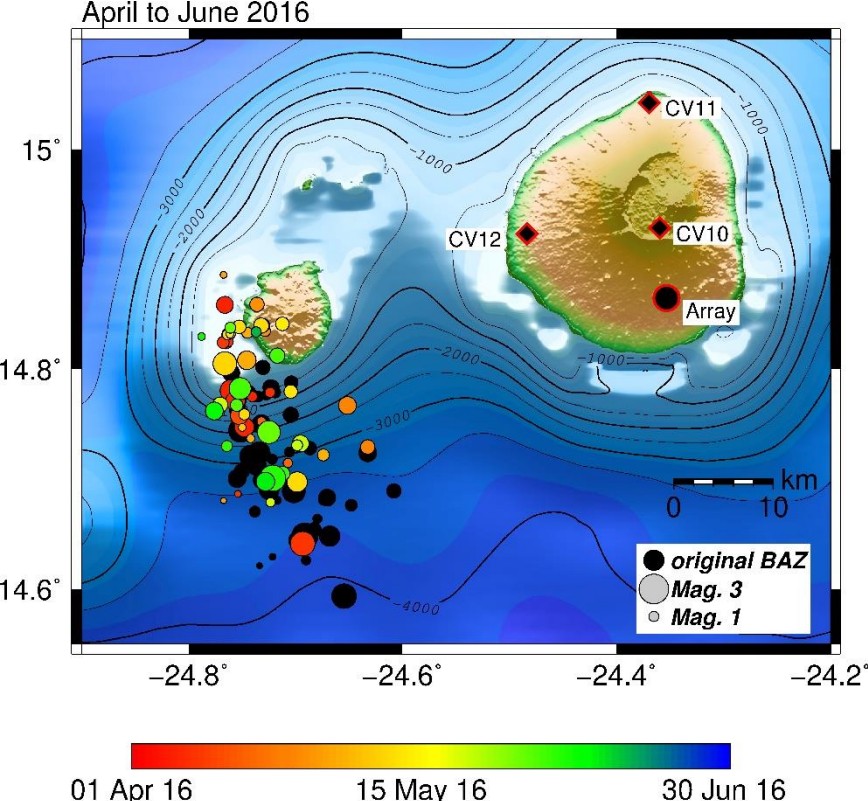

Figure 6: Earthquake locations from April to June 2016. From April to June 2016 the southernmost array station was out of operation. Comparison of events recorded in other time periods shows that the outage of this station leads to a bias of about 8.9° in backazimuth towards the south. Original locations of the earthquakes are marked by black symbols; the corrected locations (with a mean correction of 8.9° in the backazimuth determination) are marked by coloured symbols. Topographic and bathymetry data are from Ryan et al. (2009).

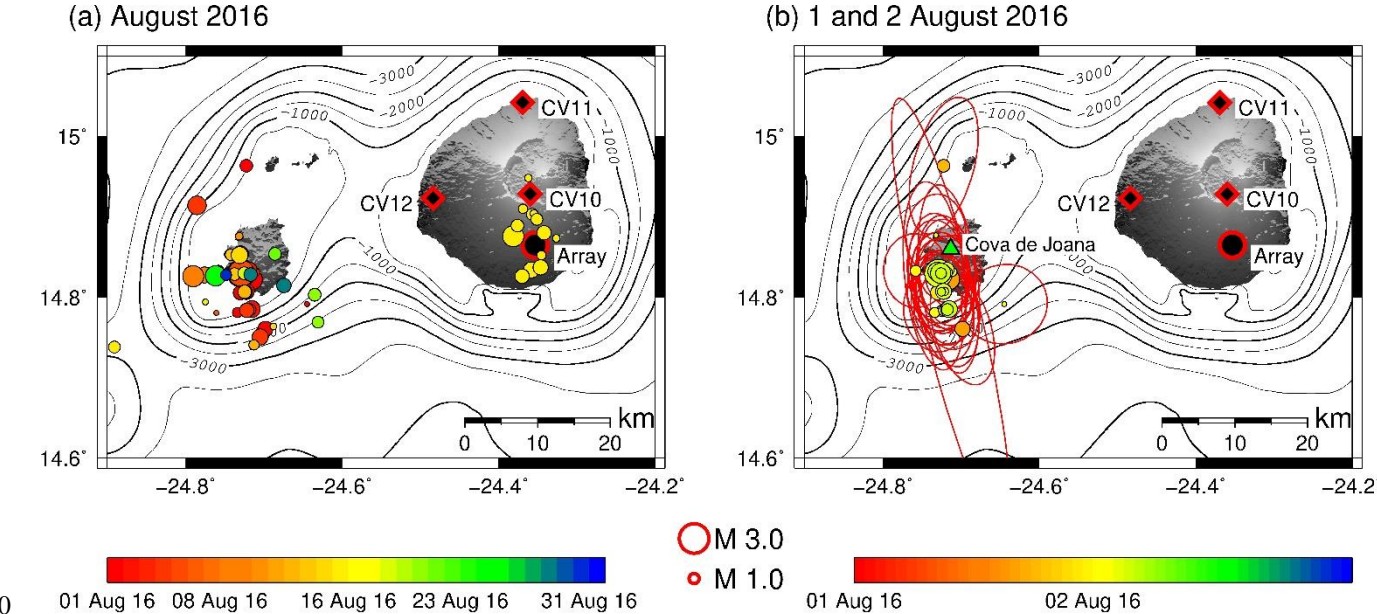

**Figure 7: (a) Earthquake locations during August 2016, including the seismic crisis. Red/black circle: position of the array, red/black diamond: additional broadband stations on Fogo. (b) Earthquake locations during the seismic crisis on 1 and 2 of August 2016. Red ellipses: errors in backazimuth and distance as determined for the array analysis. Red/black circle: position of the array, red/black diamond: additional broadband stations on Fogo. Green triangle: village Cova de Joana, evacuated during the seismic crisis. Topographic and bathymetry data are from Ryan et al. (2009).**

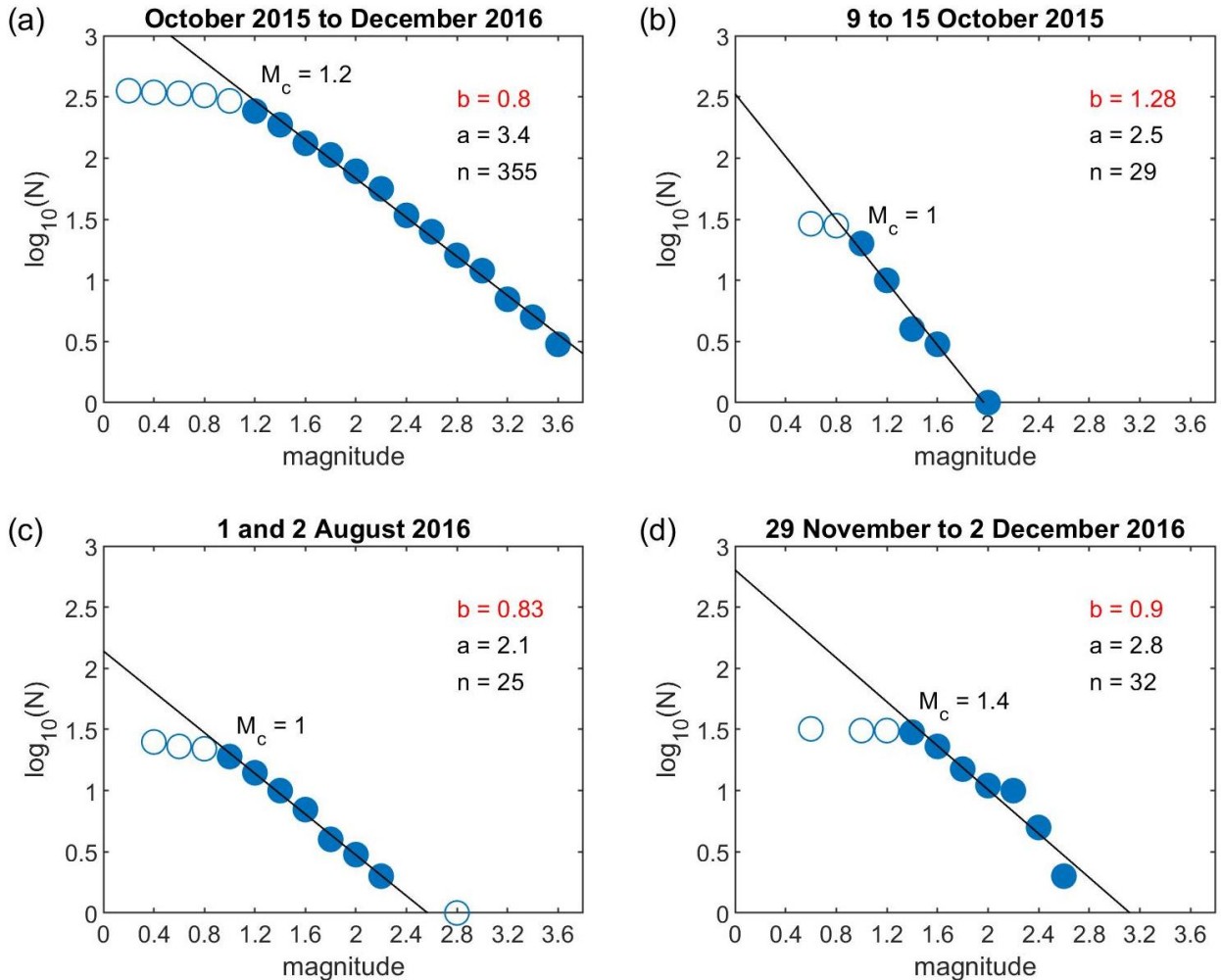

Figure 8: Magnitude–frequency relation for earthquakes observed during (a) the study period, (b) the period of elevated seismic activity from 9 to 15 October 2015, (c) the seismic crisis on 1 and 2 August 2016, and (d) the period of elevated seismic activity from 29 November to 2 December 2016. Magnitudes are binned in steps of 0.2 and $n$ corresponds to the number of events during the period under consideration. Data points used to fit the straight line for the determination of $a$ and $b$ are marked with filled dots.

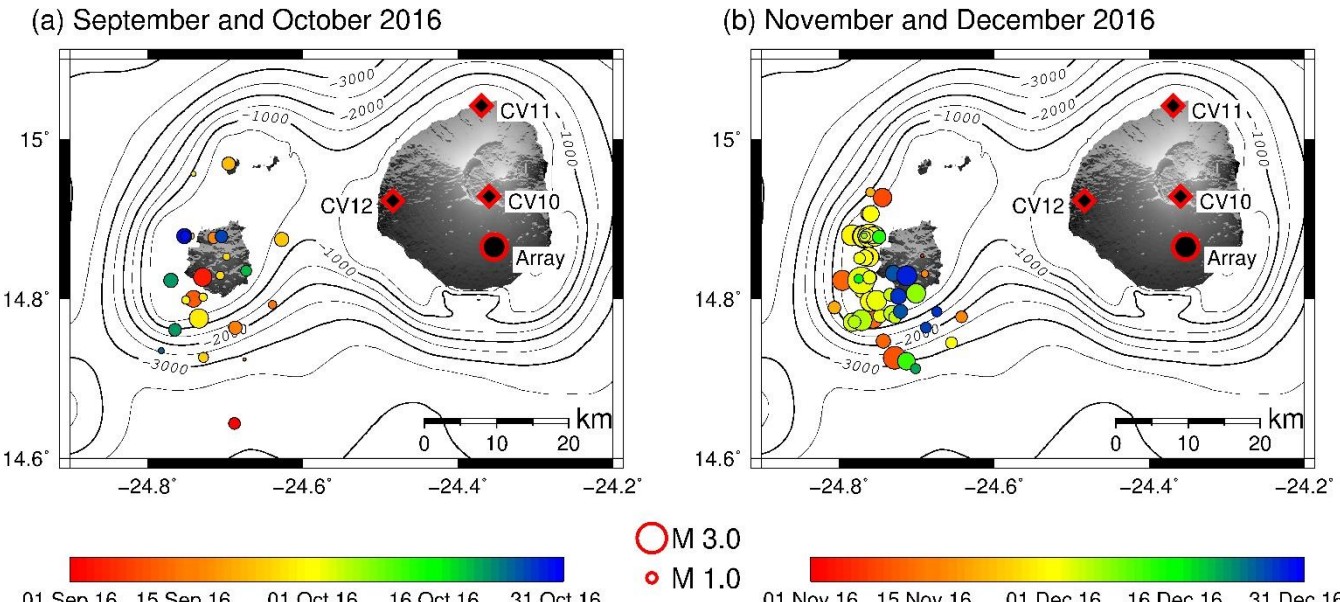

**Figure 9: (a) Earthquake locations during September and October 2016. Red/black circle: position of the array, red/black diamond:**
**additional broadband stations on Fogo. (b) The same for November and December 2016. Topographic and bathymetry data are**
**from Ryan et al. (2009).**

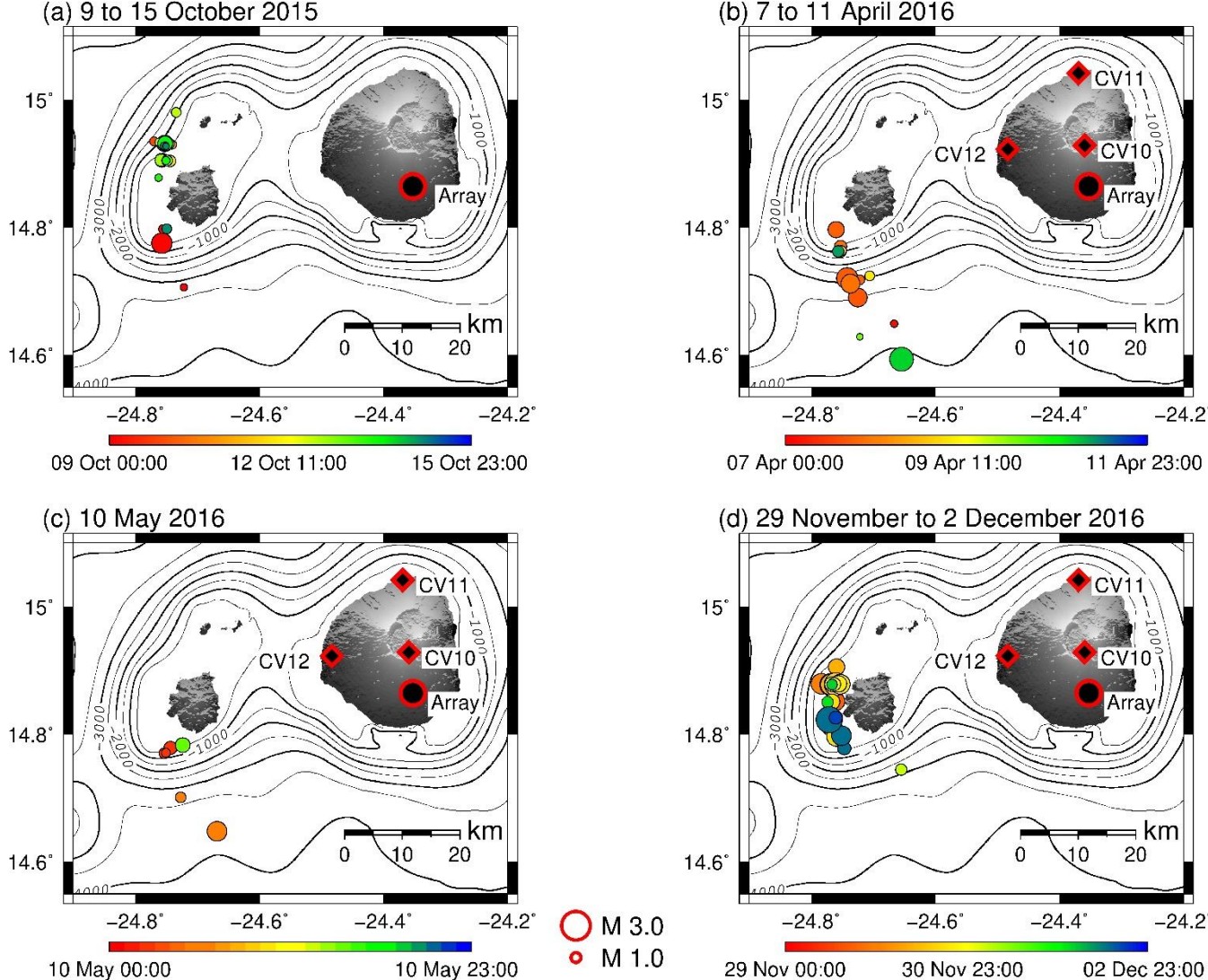

**Figure 10: Earthquake locations for four different time periods of elevated seismicity. (a) 9–15 October 2015, (b) 7–11 April 2016, (c) 10 May 2016, (d) 29 November to 2 December 2016. Red/black circle: position of the array, red/black diamonds: additional broadband stations on Fogo. Topographic and bathymetry data are from Ryan et al. (2009).**

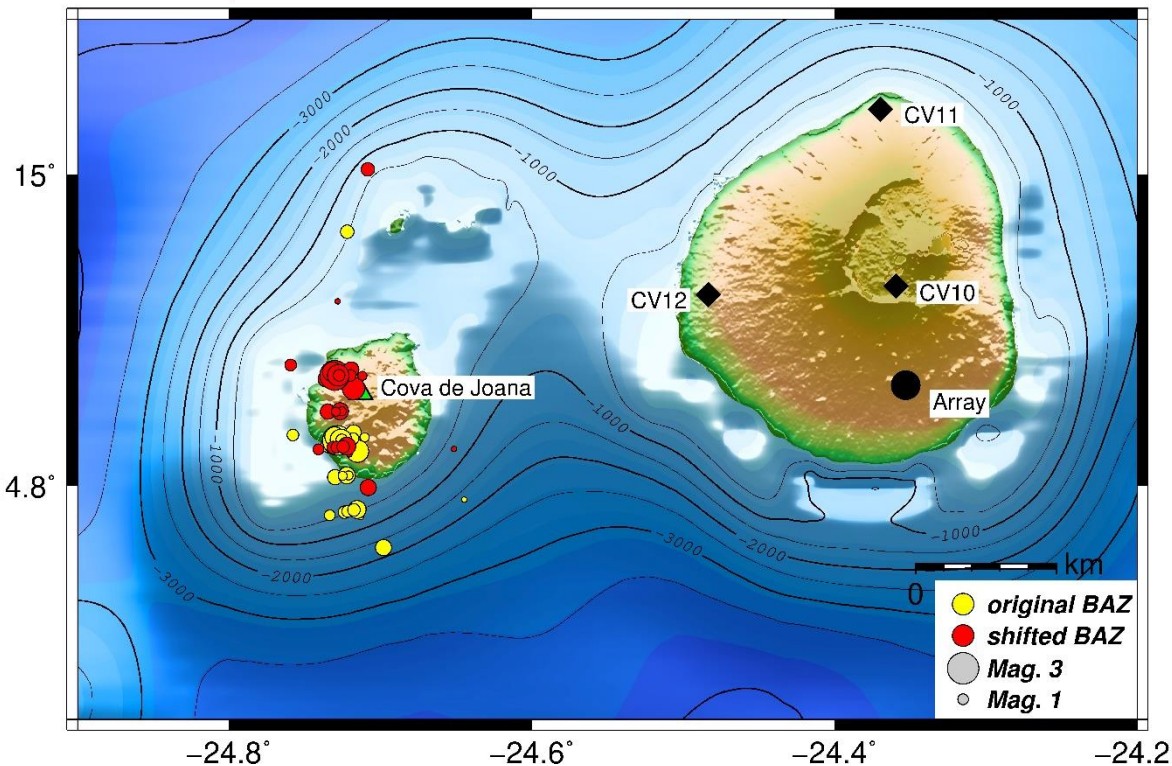

**Figure 11: Earthquake locations during the seismic crisis of 1 and 2 August 2016. Yellow circles: original location of the earthquakes determined from the array data; red circles: corrected locations according to the mean systematic deviation of 6.5° in the backazimuth determination (see text for details). Topographic and bathymetry data are from Ryan et al. (2009).**