# Peer review of "Remote monitoring of seismic swarms and the August 2016 seismic crisis of Brava, Cape Verde, using array methods"

_Natural Hazards and Earth System Sciences, 2020_

## Referee Comment (RC1) · Anonymous Referee #1 · 18 Sep 2020

The paper by Leva et al. (NHESS 2020-225), at is stage, focus on an important issue, which is the recognizing the precursors of intruding magma at crustal levels, and also the fact the Brava might be a dormant volcano, thus a contribution for the volcanic risk reduction. Despite the good approach, I have nevertheless some comments and remarks, which are the following: In line 6 it is stated that a seismic crisis occurred on Brava during the first two days of August, and in line 10 that the experiment started about only 10 month before. Which seismic baseline do you have before October 2015? Was the crisis already occurring in or before October 2015? Was the first two days just a culmination of the crisis? The total number of earthquakes mentioned in line 11 is the total recorded by the array during the experiment, including those of Fogo

and Brava, or just those of Brava? In lines 15 and 34 you pretend to show that a remote array (35 km away from the epicenters) is suitable to monitor a volcanic seismic crisis. However, in lines 155 to 157 it is mentioned the results of others authors that recorded tremors and long-period events, which the array used in this experiment wasn't able to record because it was too far away. It seems that this is a contradiction, because one of the crucial signals to be recorded in order to monitor a volcano is both long-periods events and tremors episodes. If a network/array is unable to record those signals there is no advantage to use them. The depths of hypocentres reported by Faria and Fonseca (NHESS, 2014) beneath Brava are mostly variable and there is no evidence that they are clustered at 5 km. Thus, instead of fixing the depths of all the earthquakes to 5 km (line 90), why was it not tried severals depths in order to minimize the errors ellipses, which are already quite big as suggested by the figure 5 (b). In lines 127 and 128 it is stated that "Most of the volcanic-tectonic earthquakes occurred beneath the southern part of Brava". It is most appropriate to say "located" instead of occurred, because yours locations are not so precise. What is the relevance for this paper to include the results of the paper about Fogo (lines 132-134)? Line 143 (pag. 5): please precise if the observation "…periods with elevated seismicity frequently occur beneath and around Brava." refers to the period of the experiment. If so (which seems not to be the case because your data spans only two years, or otherwise include a reference), it is more suitable to state "…periods with elevated seismicity frequently occurred beneath and around Brava during the experiment." The first phrase in line 149 (pag. 5) refers to a period during the time span by your experiment or is a general characteristic of Brava seismicity? If it is the former, please precise, otherwise include a reference. It is not clear in the first reading about the exact timing of the evolution of the seismic activity recorded on Brava during the experiment (e.g. lines 180 to 185 pag. 6). I recommend ordering it in time (and just mention it afterwards if necessary). In line 181 (pag. 6) it is stated: "… movement of the earthquake locations is related to magmatic processes.", please justify or include a reference. Distinction between offshore or around Brava (which appears in servals parts of the text) and underneath Brava must be clearer,

since a volcanic island must be seen as a whole including the submarine part of its edifice. I suggest to include in the maps a profile of the topography/bathymetry as it may help to make clearer whether the earthquakes were really offshore or (when located in the sea) on the submarine roots of the island. It is stated all along the text the terms migration, movement, shift of the seismicity. I have two observations concerning the use of those terms: 1- the uncertainties of the locations are too big (fig 5b), thus it may be that the cause of the migration/shift/movement it is just due to the random errors of the locations. 2-Examining figures 4 (a-b) and 5 (a) it seems that seismic activity was present at several places at the same time, although more intense in one zone than others. So, instead of using those terms, isn't it more suitable to say that likely (due to big errors ellipses) the seismic activity became more intense (in terms of rate ) in a certain zone than others? Final remarks: the geological setting and geotectonic of Brava were not taken in account during the discussion and/or conclusions. Why was the possibility of the movement of the faults (Madeira et al., 2010) ruled out? Or why a process of uplift episode of the island (Ramalho, 2010) was not discussed? How often the $CO_2$ fluxes measurements were done? Were they sporadic or continually? Please specify when exactly in 2016 the anomalous $CO_2$ emission was observed? Anyway I recommend a better fundamentation volcanic nature of the seismic crisis hypothesis. Why the potential Brava volcanic hazards were not included (lines 198 to 201) or mentioned in the introduction. This would reinforce the importance of the volcanic monitoring on Brava and better fit the NHESS spirit. I recommend adding a color scale to figure 1, and to make bathymetry clearer (it is hard from this figure to have an idea how the bathymetry is in vicinity of Brava is).
* * *

---

## Referee Comment (RC2) · Carmen López (Referee) · 19 Oct 2020

I find the paper by Leva et al. (2020) of great interest, since it shows how precursory volcanic activity behaves in oceanic islands; there are not many scientific papers of this type. In oceanic islands, volcanic activity monitoring involves great difficulty due to out-of-network seismic occurrence and poor network coverage, which does not facilitate the full study of the precursor phenomena. Tracking the seismic activity that accompanies the unrest is truly challenging, thus I find this paper of interest. I will now provide some recommendations and comments that I hope will be useful. Authors propose an intelligent approach, which is increasingly used in oceanic islands

and submarine volcanism, the use of seismometer arrays, which by decreasing the signal-to-noise ratio can detect low amplitude signals, even below the ambient noise. These arrays are optimal for detection, but not so good for localization, giving notable errors in azimuth and distance, especially in the case of no calibrated array and also in the case of using plane wave front approximation instead of a spherical one. Sections describing the methodology are well developed with a careful application to data and errors estimation. Array analysis was performed in the time domain, being able to locate volcano tectonic (VT) events. I wonder, if an additional analysis in the frequency domain (F-K analysis) had been carried out, whether it would have also characterized low frequency tremor or LP signals, which have not been included in the study. In fact (line 29) according to data recorded by a permanent seismic monitoring network (Faria and Day, 2017), the crisis comprised about 1000 shallow earthquakes and tremors. The localized events set their depth at 5 km, without assessing the error associated with this setting. I think other depths should be tested to know its impact on location. I think it would be desirable to get additional data, mainly about gas emissions and surface deformations, or additional seismic information for the better identification of the different stages. At this regard, it would be useful to include in Figure 3 the accumulated number of events. The variations of the "b" parameter should be discussed in more detail. During eruptive unrest phenomena, in other volcanic islands, strong variations of the "b" parameter have been observed, from values greater than 2, to close to 1, and in all cases reflecting precursory dynamic activity with swarms of VT-type events. It would also be necessary to add a figure with the temporal evolution of the "b" value. Figures show that seismicity fluctuates almost constantly, and only in certain periods is concentrated in-land, always showing dispersion. It is very possible that the dispersion is partly a product of the limitation of the array, in fact, a radial distribution of the epicenters with centre in the array is observed, showing that the semi-major axis of the error coincides with the geometry of the event cloud (fig. 6b). In this regard, if possible, it would be desirable to include the error ellipses in all locating figures (Figure 5 a, b,; Figure 6a, Figure 8 a, b, Figure 9). The
authors state that they do not observe tremor or LP signals, but the array technique used (beamforming in time) is not the best for these type of low frequency events, so I think their existence cannot be ruled out, please it can be included a clarification. I believe a further discussion about the interpretation of the phenomena is needed. The authors state "We conclude that the seismic crisis might be an example of a failed eruption, likely caused by the transport of magma and / or $CO_2$ into the upper crust, as it has been suggested by the observed changes on diffuse $CO_2$ degassing surveys ", lines 230-232. To state that, it would be necessary to analyse results with data from local monitoring networks, including gas emission and, if it was the case, deformation, occurring during the studied period. In addition, an interpretation based on the knowledge of the structure and the geological frame would be recommended.

Please also note the supplement to this comment:
https://nhess.copernicus.org/preprints/nhess-2020-225/nhess-2020-225-RC2-supplement.pdf

**Supplement:**

[revised manuscript text omitted]

---

## Author Comment (AC1) · 22 Oct 2020

Response to the interactive comment on "Remote monitoring of seismic swarms and the August 2016 seismic crisis of Brava, Cape Verde, using array methods" by Carola Leva et al.

Anonymous Referee #1

We would like to thank the reviewer for the helpful comments and constructive suggestions.

R1: The paper by Leva et al. (NHESS 2020-225), at is stage, focus on an important

issue, which is the recognizing the precursors of intruding magma at crustal levels, and also the fact the Brava might be a dormant volcano, thus a contribution for the volcanic risk reduction. Despite the good approach, I have nevertheless some comments and remarks, which are the following: In line 6 it is stated that a seismic crisis occurred on Brava during the first two days of August, and in line 10 that the experiment started about only 10 month before. Which seismic baseline do you have before October 2015? Was the crisis already occurring in or before October 2015? Was the first two days just a culmination of the crisis?

A.: This is a very good point and we will include it in the discussion. We do not have access to data before October 2015, thus our baseline starts in October 2015 and we cannot comment on the seismicity before our study. However, Faria and Day (2017) state that the seismicity from 2011 to 2015 showed a constant rate with "sporadic peaks" and changed after an earthquake with magnitude M4 in September 2015. We will include this information in the discussion.

R1: The total number of earthquakes mentioned in line 11 is the total recorded by the array during the experiment, including those of Fogo and Brava, or just those of Brava?

A.: This is the total number of local earthquakes recorded by the network, including earthquakes of Fogo and Brava.

R1: In lines 15 and 34 you pretend to show that a remote array (35 km away from the epicenters) is suitable to monitor a volcanic seismic crisis. However, in lines 155 to 157 it is mentioned the results of others authors that recorded tremors and long-period events, which the array used in this experiment wasn't able to record because it was too far away. It seems that this is a contradiction, because one of the crucial signals to be recorded in order to monitor a volcano is both long-periods events and tremors episodes. If a network/array is unable to record those signals there is no advantage to use them.

A.: A local network, of course, can provide further (sometimes more) information. However, without the remote array we would not have any information about the seismic crisis on Brava and that is an obvious advantage. We show how this data can be used to gain as much relevant information as possible.

R1: The depths of hypocentres reported by Faria and Fonseca (NHESS, 2014) beneath Brava are mostly variable and there is no evidence that they are clustered at 5 km. Thus, instead of fixing the depths of all the earthquakes to 5 km (line 90), why was it not tried severals depths in order to minimize the errors ellipses, which are already quite big as suggested by the figure 5 (b).

A.: We performed a careful analysis of the contributions to the error of the epicentral distance by evaluating the influence of all parameters used for the distance estimation. It turned out, that a variation of the event depth only has a minor impact on the result, compared to other variables (lines 91, 94-97). After this error analysis we found that an error of 10% for the distance in general covers best the errors resulting from the uncertainties of the distance estimation (lines 97/98). We decided to use this relative conservative estimate for the error to incorporate the uncertainties of the simple two–layer assumption, including the uncertainties of the depth. This is already described in the text, but we will clarify this point during revision.

R1: In lines 127 and 128 it is stated that "Most of the volcanic-tectonic earthquakes occurred beneath the southern part of Brava". It is most appropriate to say "located" instead of occurred, because yours locations are not so precise.

A.: Thank you for pointing this out, we will modify this in the revised manuscript.

R1: What is the relevance for this paper to include the results of the paper about Fogo (lines 132-134)?

A.: We include the results here, because the earthquakes beneath Fogo are a rare observation and this information helps to provide a more complete image of seismicity in the region, which can be seen when comparing Figures 5, 6, 8 and 9.

R1: Line 143 (pag. 5): please precise if the observation ". . .periods with elevated seismicity frequently occur beneath and around Brava." refers to the period of the experiment. If so (which seems not to be the case because your data spans only two years, or otherwise include a reference), it is more suitable to state ". . .periods with elevated seismicity frequently occurred beneath and around Brava during the experiment."

A.: Thank you for the suggestion, we will modify this statement in the revised manuscript.

R1: The first phrase in line 149 (pag. 5) refers to a period during the time span by your experiment or is a general characteristic of Brava seismicity? If it is the former, please precise, otherwise include a reference.

A.: Thank you for pointing this out. Both is the case, the seismicity is characterized by this shift, which we observe. Comparing the earthquake locations from former studies, this feature is confirmed. We will precise this statement and include the references.

R1: It is not clear in the first reading about the exact timing of the evolution of the seismic activity recorded on Brava during the experiment (e.g. lines 180 to 185 pag. 6). I recommend ordering it in time (and just mention it afterwards if necessary).

A.: We will change the order of the description in the revised manuscript.

R1: In line 181 (pag. 6) it is stated: ". . . movement of the earthquake locations is related to magmatic processes.", please justify or include a reference.

A.: As suggested, we will include a reference in the manuscript.

R1: Distinction between offshore or around Brava (which appears in servals parts of the text) and underneath Brava must be clearer, since a volcanic island must be seen as a whole including the submarine part of its edifice. I suggest to include in the maps a profile of the topography/bathymetry as it may help to make clearer whether the earthquakes were really offshore or (when located in the sea) on the submarine roots

of the island.

A.: As suggested, we will add additional contour lines to the maps shown.

R1: It is stated all along the text the terms migration, movement, shift of the seismicity. I have two observations concerning the use of those terms: 1- the uncertainties of the locations are too big (fig 5b), thus it may be that the cause of the migration/shift/movement it is just due to the random errors of the locations. 2-Examining figures 4 (a-b) and 5 (a) it seems that seismic activity was present at several places at the same time, although more intense in one zone than others. So, instead of using those terms, isn't it more suitable to say that likely (due to big errors ellipses) the seismic activity became more intense (in terms of rate) in a certain zone than others?

A.: While our observations cannot constrain individual earthquake locations exactly, we can still detect systematic shifts in seismic activity, even if random errors are taken into account (as we have done). From our observations, we cannot confirm (nor fully exclude) that there is a continuous wide-spread low-level activity in the entire region. We therefore prefer to describe our observations by "shift" rather than by "variations of intensity". But we agree that "migration" or "movement" may be less appropriate, as this may give the impression that events are directly related (as if aligned along a common fault, which is probably not the case here). We will modify the expressions in the revised version.

R1: Final remarks: the geological setting and geotectonic of Brava were not taken in account during the discussion and/or conclusions. Why was the possibility of the movement of the faults (Madeira et al., 2010) ruled out?

A.: We cannot completely rule out this possibility. However, for a comment on the link between the earthquakes and the faults, we would need a precise location in addition to focal mechanisms of the earthquake. Being unable to determine the depth makes this even more difficult. However, we can suggest a possible magmatic origin, as the earthquakes occur in swarms and not in a mainshock-aftershock sequence, which

would be expected for tectonic events.

R1: Or why a process of uplift episode of the island (Ramalho, 2010) was not discussed?

A.: Ramalho state that Brava has experienced significant uplift, which cannot be explained by a regional uplift across the swell, but rather by a local uplift. The cause of the uplift could e.g. be the magmatic intrusion below the edifice. A failed eruption could contribute to such an uplift, however we cannot comment on the amount of material added and thus on a potential uplift. Taken together our observations of 2016 and the observation of Faria and Day (2017), the seismicity in 2016 could indeed be part of an uplift episode. We will include the reference and extend the discussion accordingly.

R1: How often the $CO_2$ fluxes measurements were done? Were they sporadic or continually? Please specify when exactly in 2016 the anomalous $CO_2$ emission was observed?

A.: The $CO_2$ emission surveys were carried out every 2 years since 2010 (see references). There were two measurements taken in 2016, one in August and one in October/ November. The measurement of October/ November shows the highest values measured since 2010. The reason for the background level values of August is most likely the timing of the survey. In August the rainy season distorts the $CO_2$ emission measurements. Therefore, the surveys in the years before 2016 were taken outside of the rainy season, making it difficult to compare the August 2016 data to the data of previous years. The data of October/ November 2016 however are comparable to previous measurements and thus more meaningful (Pérez 2020, personal communication). The survey of 2018 showed lower levels of $CO_2$ emissions again. However, for details we have to refer the reader to the cited references.

R1: Anyway I recommend a better fundamentation volcanic nature of the seismic crisis hypothesis. Why the potential Brava volcanic hazards were not included (lines 198 to 201) or mentioned in the introduction. This would reinforce the importance of the

volcanic monitoring on Brava and better fit the NHESS spirit.

A.: Thank you for pointing this out, we will adjust the discussion accordingly.

R1: I recommend adding a color scale to figure 1, and to make bathymetry clearer (it is hard from this figure to have an idea how the bathymetry is in vicinity of Brava is).

A.: We will add contour lines to the map, as also recommended for the other maps.

―――――――――――――――――――――

---

## Author Comment (AC2) · 22 Oct 2020

Response to the interactive comment on "Remote monitoring of seismic swarms and the August 2016 seismic crisis of Brava, Cape Verde, using array methods" by Carola Leva et al.

Reviewer 2 - Carmen López

We would like to thank the reviewer for the helpful comments and constructive suggestions.

R2: I find the paper by Leva et al. (2020) of great interest, since it shows how precur-

sory volcanic activity behaves in oceanic islands; there are not many scientific papers of this type. In oceanic islands, volcanic activity monitoring involves great difficulty due to out-of-network seismic occurrence and poor network coverage, which does not facilitate the full study of the precursor phenomena. Tracking the seismic activity that accompanies the unrest is truly challenging, thus I find this paper of interest. I will now provide some recommendations and comments that I hope will be useful. Authors propose an intelligent approach, which is increasingly used in oceanic islands and submarine volcanism, the use of seismometer arrays, which by decreasing the signal-to-noise ratio can detect low amplitude signals, even below the ambient noise. These arrays are optimal for detection, but not so good for localization, giving notable errors in azimuth and distance, especially in the case of no calibrated array and also in the case of using plane wave front approximation instead of a spherical one. Sections describing the methodology are well developed with a careful application to data and errors estimation. Array analysis was performed in the time domain, being able to locate volcano tectonic (VT) events. I wonder, if an additional analysis in the frequency domain (F-K analysis) had been carried out, whether it would have also characterized low frequency tremor or LP signals, which have not been included in the study. In fact (line 29) according to data recorded by a permanent seismic monitoring network (Faria and Day, 2017), the crisis comprised about 1000 shallow earthquakes and tremors.

A.: We thank the reviewer for the appreciation of our work. The tremor signals produce smaller amplitudes, which are likely suppressed by noise in the distance of the array of 35 km. We did not carry out a F-K analysis, but could not find any indication for such signals by manual inspection of the seismograms using different filters. Additionally, we applied different sta-/lta-triggers to detect events of different frequency content and could not find any tremors or long-period events that originated on Brava, especially during the seismic crisis in August 2016.

R2: The localized events set their depth at 5 km, without assessing the error associated with this setting. I think other depths should be tested to know its impact on location.

A.: We performed a careful analysis of the contributions to the error of the epicentral distance by evaluating the influence of all parameters used for the distance estimation. It turned out, that a variation of the event depth only has a minor impact on the result, compared to other variables (lines 91, 94-97). After this error analysis we found that an error of 10% for the distance in general covers best the errors resulting from the uncertainties of the distance estimation (lines 97/98). We decided to use this relative conservative estimate for the error to incorporate the uncertainties of the simple two–layer assumption, including the uncertainties of the depth. This is already described in the text, but we will clarify this point during revision.

R2: I think it would be desirable to get additional data, mainly about gas emissions and surface deformations, or additional seismic information for the better identification of the different stages. A.: Yes, we agree that it would be desirable to have additional data. To our knowledge, there are other groups working on a publication about gas emission data. Unfortunately, the data is not available to us. The seismic data of the local monitoring network is also restricted.

R2: At this regard, it would be useful to include in Figure 3 the accumulated number of events.

A.: Thank you for the suggestion, we will modify the Figure accordingly.

R2: The variations of the "b" parameter should be discussed in more detail. During eruptive unrest phenomena, in other volcanic islands, strong variations of the "b" parameter have been observed, from values greater than 2, to close to 1, and in all cases reflecting precursory dynamic activity with swarms of VT-type events. It would also be necessary to add a figure with the temporal evolution of the "b" value.

A.: Thank you for pointing this out. It is difficult to assess the precise b-value as we deal with rather low numbers of events (line 129). However, we will clarify how we estimate the b-value and add a figure of the b-value for the complete study period. We have also looked at the b-value variation within 3 month intervals and will add the corresponding

figures to the supplementary material. For a more detailed interpretation we would need longer observation times of several years.

R2: Figures show that seismicity fluctuates almost constantly, and only in certain periods is concentrated in-land, always showing dispersion. It is very possible that the dispersion is partly a product of the limitation of the array, in fact, a radial distribution of the epicenters with centre in the array is observed, showing that the semi-major axis of the error coincides with the geometry of the event cloud (fig. 6b). In this regard, if possible, it would be desirable to include the error ellipses in all locating figures (Figure 5 a, b,; Figure 6a, Figure 8 a, b, Figure 9).

A.: This is only apparently the case, in other months this is not observed. Possibly this apparent dispersion could, under consideration of the error of the backazimuth, be interpreted as an indicator, that the events cluster more closely. Nevertheless, we observe a relative shift of the event locations over the study period. A detailed analysis shows that there is a systematic difference in events west and south of Brava, which cannot be explained by a random error in BAZ. We decided to not include the error ellipses in all figures, as this strongly influences the readability of the maps. Nevertheless, we will add a figure with the error ellipses to the supplementary material.

R2: The authors state that they do not observe tremor or LP signals, but the array technique used (beamforming in time) is not the best for these type of low frequency events, so I think their existence cannot be ruled out, please it can be included a clarification.

A.: Please refer to our response to an earlier question above. We do not rule out the existence of different event types that could be a precursor of the crisis, we suggest that their absence in our data can be explained by the rather large distance of 35 km between array and possible source locations near Brava (line 166). We do not observe other event types originating from Brava. From the earthquake analysis we also do not find precursors. However, we will clarify this in the revised manuscript.

R2: I believe a further discussion about the interpretation of the phenomena is needed. The authors state "We conclude that the seismic crisis might be an example of a failed eruption, likely caused by the transport of magma and / or CO2 into the upper crust, as it has been suggested by the observed changes on diffuse CO2 degassing surveys ", lines 230-232. To state that, it would be necessary to analyse results with data from local monitoring networks, including gas emission and, if it was the case, deformation, occurring during the studied period. In addition, an interpretation based on the knowledge of the structure and the geological frame would be recommended.

A.: Yes, we agree that it would be interesting to directly compare the gas emission data and the data of the local monitoring network to our data. Unfortunately, they are not available to us. In the revised manuscript we have included a discussion about a possible uplift period in 2016. However, more data, especially of a possible deformation, would be desirable. We will include an outlook in the conclusions, pointing out the necessity of including information from other disciplines to better assess volcanic hazards.

Additional comments taken from the annotated manuscript

Line 90: I do not understand why all events set their depth at 5 km. I think other depths should be tested to know its impact, and select which one minimize errors.

A.: We tested the impact of different depths (and different crustal and mantle velocities as well as different Moho depths). We understand, that our description of this error analysis (line 94-100) might be misleading and we will clarify this point during revision.

Line 112: In my understanding, the periods referred to from here on, are not presented month by month. And I have some difficulties in understand the reasons for time periods selection. Please clarify the distinctive characteristics of each one of them.

A.: Thank you for pointing this out. We described the periods with elevated seismicity for each month. We will clarify this in the revised manuscript.

[Figure]

Line 136: Why activity in this period is not considered as a seismic crisis? between 29-30 November you have even more events that in previous periods.

A.: The term seismic crisis referred to the period with elevated seismicity beneath Brava, leading to evacuation of a village on Brava. The increased activity from 29 November to 2 December occurred offshore and the alert level for Brava was not raised.

---

## Author Response (AR1)

**Authors response**
**"Remote monitoring of seismic swarms and the August 2016 seismic crisis of Brava, Cape Verde, using array methods" by Carola Leva et al.**

Referee 1: The paper by Leva et al. (NHESS 2020-225), at is stage, focus on an important issue, which is the recognizing the precursors of intruding magma at crustal levels, and also the fact the Brava might be a dormant volcano, thus a contribution for the volcanic risk reduction. Despite the good approach, I have nevertheless some comments and remarks, which are the following: In line 6 it is stated that a seismic crisis occurred on Brava during the first two days of August, and in line 10 that the experiment started about only 10 month before. Which seismic baseline do you have before October 2015? Was the crisis already occurring in or before October 2015? Was the first two days just a culmination of the crisis?

*Answer: We thank the reviewer for the appreciation of our work. This is a very good point. We do not have access to data before October 2015, thus our baseline starts in October 2015 and we cannot comment on the seismicity before our study. However, Faria and Day (2017) state that the seismicity from 2011 to 2015 showed a constant rate with "sporadic peaks" and changed after an earthquake with magnitude M4 in September 2015. We refer to this observation in the discussion (lines 201-202).*

R1: The total number of earthquakes mentioned in line 11 is the total recorded by the array during the experiment, including those of Fogo and Brava, or just those of Brava?

*A.: This is the total number of local earthquakes recorded by the network, including earthquakes of Fogo and Brava. We modified the text for clarification in line 11.*

R1: In lines 15 and 34 you pretend to show that a remote array (35 km away from the epicenters) is suitable to monitor a volcanic seismic crisis. However, in lines 155 to 157 it is mentioned the results of others authors that recorded tremors and long-period events, which the array used in this experiment wasn't able to record because it was too far away. It seems that this is a contradiction, because one of the crucial signals to be recorded in order to monitor a volcano is both long-periods events and tremors episodes. If a network/array is unable to record those signals there is no advantage to use them.

*A.: We agree that a local network, of course, can provide further information. However, without the remote array we would not have any information about the seismic crisis on Brava and that is an obvious advantage. We show how this data can be used to gain as much relevant information as possible.*

R1: The depths of hypocentres reported by Faria and Fonseca (NHESS, 2014) beneath Brava are mostly variable and there is no evidence that they are clustered at 5 km. Thus, instead of fixing the depths of all the earthquakes to 5 km (line 90), why was it not tried severals depths in order to minimize the errors ellipses, which are already quite big as suggested by the figure 5 (b).

*A.: We performed a careful analysis of the different contributions to the error of the epicentral distance estimation. It turned out, that a variation of the event depth only has a minor impact on the result, compared to other parameters such as crustal and upper-mantle velocities, for example. We found that an error of 10% for the distance in general covers best the errors resulting from the uncertainties of the distance estimation. We decided to use this relative conservative estimate for the error to incorporate the uncertainties of the simple two–layer assumption, including the uncertainties of the depth. We modified the text accordingly (lines 97-101).*

R1: In lines 127 and 128 it is stated that "Most of the volcanic-tectonic earthquakes occurred beneath the southern part of Brava". It is most appropriate to say "located" instead of occurred, because yours locations are not so precise.

*A.: Thank you for pointing this out, we modified the text accordingly (line 135).*

R1: What is the relevance for this paper to include the results of the paper about Fogo (lines 132-134)?

*A.: We include the results here, because the earthquakes beneath Fogo are a rare observation and this information helps to provide a more complete image of seismicity in the region, which can be seen when comparing Figures 5, 7 and 9.*

R1: Line 143 (pag. 5): please precise if the observation ". . .periods with elevated seismicity frequently occur beneath and around Brava." refers to the period of the experiment. If so (which seems not to be the case because your data spans only two years, or otherwise include a reference), it is more suitable to state ". . .periods with elevated seismicity frequently occurred beneath and around Brava during the experiment."

*A.: Thank you for the suggestion, we modified the statement in the revised manuscript (line 158).*

R1: The first phrase in line 149 (pag. 5) refers to a period during the time span by your experiment or is a general characteristic of Brava seismicity? If it is the former, please precise, otherwise include a reference.

*A.: Thank you for pointing this out. Both is true, the seismicity is characterized by this shift, which we observe. Comparing the earthquake locations from former studies, this feature is confirmed. We specified this statement and included the references (lines 165-167).*

R1: It is not clear in the first reading about the exact timing of the evolution of the seismic activity recorded on Brava during the experiment (e.g. lines 180 to 185 pag. 6). I recommend ordering it in time (and just mention it afterwards if necessary).

*A.: We changed the order of the description in lines 205-207.*

R1: In line 181 (pag. 6) it is stated: ". . . movement of the earthquake locations is related to magmatic processes.", please justify or include a reference.

*A.: As suggested, we included the references in the manuscript (line 212-215).*

R1: Distinction between offshore or around Brava (which appears in servals parts of the text) and underneath Brava must be clearer, since a volcanic island must be seen as a whole including the submarine part of its edifice. I suggest to include in the maps a profile of the topography/bathymetry as it may help to make clearer whether the earthquakes were really offshore or (when located in the sea) on the submarine roots of the island.

*A.: As suggested, we added additional contour lines to the maps shown.*

R1: It is stated all along the text the terms migration, movement, shift of the seismicity. I have two observations concerning the use of those terms: 1- the uncertainties of the locations are too big (fig 5b), thus it may be that the cause of the migration/shift/movement it is just due to the random errors of the locations. 2-Examining figures 4 (a-b) and 5 (a) it seems that seismic activity was present at several places at the same time, although more intense in one zone than others. So, instead of using those terms, isn't it more suitable to say that likely (due to big errors ellipses) the seismic activity became more intense (in terms of rate) in a certain zone than others?

*A.: While our observations cannot constrain individual earthquake locations exactly, we can still detect systematic shifts in seismic activity, even if random errors are taken into account (as we have done). From our observations, we cannot confirm (nor fully exclude) that there is a continuous wide-spread low-level activity in the entire region. We agree that "migration" or "movement" may be less appropriate, as this may give the impression that events are moving or directly related (as if aligned along a common fault, which is probably not the case here). We modified the expressions in the revised version.*

R1: Final remarks: the geological setting and geotectonic of Brava were not taken in account during the discussion and/or conclusions. Why was the possibility of the movement of the faults (Madeira et al., 2010) ruled out?

*A.: We cannot completely rule out this possibility. However, for a comment on the link between the earthquakes and the faults, we would need more precise locations in addition to focal mechanisms of the earthquakes. Being unable to determine the depth makes this even more difficult. However, we can suggest a possible magmatic origin, as the earthquakes occur in swarms and not in a mainshock-aftershock sequence, which would be expected for tectonic events. Nevertheless, we included this point in the discussion (lines 246-251).*

R1: Or why a process of uplift episode of the island (Ramalho, 2010) was not discussed?

*A.: Ramalho state that Brava has experienced significant uplift, which cannot be explained by a regional uplift across the swell, but rather by a local source of uplift. The cause of the uplift could e.g. be the magmatic intrusion below the edifice. A failed eruption could contribute to such an uplift, however we cannot comment on the amount of material added and thus on a potential uplift. Taken together our observations of 2016 and the observation of Faria and Day (2017), the seismicity in 2016 could indeed be part of an uplift episode. We included the reference and extended the discussion accordingly (line 240-245).*

R1: How often the CO2 fluxes measurements were done? Were they sporadic or continually? Please specify when exactly in 2016 the anomalous CO2 emission was observed?

*A.: The CO2 emission surveys were carried out every 2 years since 2010 (see references). There were two measurement campaigns in 2016, one in August and one in October/ November. The measurement of October/ November shows the highest values measured since 2010. The reason for the background level values of August is most likely the timing of the survey. In August the rainy season distorts the CO2 emission measurements. Therefore, the surveys in the years before 2016 were taken outside of the rainy season, making it difficult to compare the August 2016 data to the data of previous years. The data of October/ November 2016 however are comparable to previous measurements and thus more*

*meaningful (Pérez 2020, personal communication). The survey of 2018 showed lower levels of CO2 emissions again. However, for details we have to refer the reader to the cited references.*

R1: Anyway I recommend a better fundamentation volcanic nature of the seismic crisis hypothesis. Why the potential Brava volcanic hazards were not included (lines 198 to 201) or mentioned in the introduction. This would reinforce the importance of the volcanic monitoring on Brava and better fit the NHESS spirit.

*A.: Thank you for pointing this out, we adjusted the introduction and discussion accordingly (line 22-23, 225-227).*

R1: I recommend adding a color scale to figure 1, and to make bathymetry clearer (it is hard from this figure to have an idea how the bathymetry is in vicinity of Brava is).

*A.: We added contour lines to the map, as also recommended for the other maps.*

Referee 2 (Carmen López): I find the paper by Leva et al. (2020) of great interest, since it shows how precursory volcanic activity behaves in oceanic islands; there are not many scientific papers of this type. In oceanic islands, volcanic activity monitoring involves great difficulty due to out-of-network seismic occurrence and poor network coverage, which does not facilitate the full study of the precursor phenomena. Tracking the seismic activity that accompanies the unrest is truly challenging, thus I find this paper of interest. I will now provide some recommendations and comments that I hope will be useful. Authors propose an intelligent approach, which is increasingly used in oceanic islands and submarine volcanism, the use of seismometer arrays, which by decreasing the signal-to-noise ratio can detect low amplitude signals, even below the ambient noise. These arrays are optimal for detection, but not so good for localization, giving notable errors in azimuth and distance, especially in the case of no calibrated array and also in the case of using plane wave front approximation instead of a spherical one. Sections describing the methodology are well developed with a careful application to data and errors estimation. Array analysis was performed in the time domain, being able to locate volcano tectonic (VT) events. I wonder, if an additional analysis in the frequency domain (F-K analysis) had been carried out, whether it would have also characterized low frequency tremor or LP signals, which have not been included in the study. In fact (line 29) according to data recorded by a permanent seismic monitoring network (Faria and Day, 2017), the crisis comprised about 1000 shallow earthquakes and tremors.

*Answer: We thank the reviewer for the appreciation of our work. The tremor signals produce smaller amplitudes, which are likely suppressed by noise in the distance of the array of 35 km. We did not carry out a F-K analysis, but could not find any indication for tremor signals by manual inspection of the seismograms using different filters. Additionally, we applied different sta-/lta-triggers to detect events of different frequency content and could not find any tremors or long-period events that originated on Brava, especially during the seismic crisis in August 2016.*

R2: The localized events set their depth at 5 km, without assessing the error associated with this setting. I think other depths should be tested to know its impact on location.

*A.: We performed a careful analysis of the different contributions to the error of the epicentral distance estimation. It turned out, that a variation of the event depth only has a minor impact on the result, compared to other parameters such as crustal and upper-mantle velocities, for example. We found that an error of 10% for the distance in general covers best the errors resulting from the uncertainties of the distance estimation. We decided to use this relative conservative estimate for the error to incorporate the uncertainties of the simple two–layer assumption, including the uncertainties of the depth. We modified the text accordingly (lines 97-101).*

R2: I think it would be desirable to get additional data, mainly about gas emissions and surface deformations, or additional seismic information for the better identification of the different stages.

*A.: Yes, we agree that it would be desirable to have additional data. To our knowledge, there are other groups working on a publication based on gas-emission data. Unfortunately, the data is not available to us. The seismic data of the local monitoring network is also restricted.*

R2: At this regard, it would be useful to include in Figure 3 the accumulated number of events.

*A.: Thank you for the suggestion, we modified Figure 3 accordingly.*

R2: The variations of the "b" parameter should be discussed in more detail. During eruptive unrest phenomena, in other volcanic islands, strong variations of the "b" parameter have been observed, from values greater than 2, to close to 1, and in all cases reflecting precursory dynamic activity with swarms of VT-type events. It would also be necessary to add a figure with the temporal evolution of the "b" value.

*A.: Thank you for pointing this out. It is difficult to assess the precise b-value as we deal with rather small numbers of events (line 135). However, we clarified how we estimate the b-value (line 136-141) and added a figure of the b-value for the complete study period. We have also looked at the b-value variation within 3-months intervals and added the corresponding figures to the supplementary material. For a more detailed interpretation longer observation times of several years are required. During the revision we performed several tests to estimate the reliability of the b-values and decided to include further comments on the uncertainty of their determination (lines 141-144).*

R2: Figures show that seismicity fluctuates almost constantly, and only in certain periods is concentrated in-land, always showing dispersion. It is very possible that the dispersion is partly a product of the limitation of the array, in fact, a radial distribution of the epicenters with centre in the array is observed, showing that the semi-major axis of the error coincides with the geometry of the event cloud (fig. 6b). In this regard, if possible, it would be desirable to include the error ellipses in all locating figures (Figure 5 a, b,; Figure 6a, Figure 8 a, b, Figure 9).

*A.: This is only apparently the case, in other months this is not observed. Possibly this apparent dispersion could, under consideration of the error of the backazimuth, be interpreted as an indicator, that the events cluster more closely. Nevertheless, we observe a relative shift of the event locations over the study period. A detailed analysis shows that there is a systematic difference in events west and south of Brava, which cannot be explained by a random error in BAZ.*
*We decided to not include the error ellipses in all figures, as this strongly influences the readability of the maps. Nevertheless, we added a figure with the error ellipses to the supplementary material.*

R2: The authors state that they do not observe tremor or LP signals, but the array technique used (beamforming in time) is not the best for these type of low frequency events, so I think their existence cannot be ruled out, please it can be included a clarification.

*A.: Please refer to our response to an earlier question above. We do not rule out the existence of different event types that could be a precursor of the crisis, we suggest that their absence in our data can be explained by the large distance of 35 km between array and possible source locations near Brava. We do not observe other event types originating from Brava. From the earthquake analysis we also do not find precursors. However, we clarified this in the revised manuscript (line 175-176).*

R2: I believe a further discussion about the interpretation of the phenomena is needed. The authors state "We conclude that the seismic crisis might be an example of a failed eruption, likely caused by the transport of magma and / or CO2 into the upper crust, as it has been suggested by the observed changes on diffuse CO2 degassing surveys ", lines 230-232. To state that, it would be necessary to analyse results with data from local monitoring networks, including gas emission and, if it was the case, deformation, occurring during the studied period. In addition, an interpretation based on the knowledge of the structure and the geological frame would be recommended.

*A.: Yes, we agree that it would be interesting to directly compare the gas-emission data and the data of the local monitoring network to our data. Unfortunately, they are not available to us. In the revised manuscript we have included a discussion about a possible uplift period in 2016 (lines 240–245). However, more data, e.g. on crustal deformation, would be desirable. We included an outlook in the conclusions, pointing out the necessity of including information from other disciplines to better assess volcanic hazards (lines 271–273).*

**Additional comments taken from the annotated manuscript**

Line 90: I do not understand why all events set their depth at 5 km. I think other depths should be tested to know its impact, and select which one minimize errors.

*A.: We tested the impact of different depths (and different crustal and upper-mantle velocities as well as different Moho depths). We understand, that our description of this error analysis might have been misleading and we clarified this point (lines 97–101).*

Line 112: In my understanding, the periods referred to from here on, are not presented month by month. And I have some difficulties in understand the reasons for time periods selection. Please clarify the distinctive characteristics of each one of them.

*A.: Thank you for pointing this out. We described the periods with elevated seismicity for each month. We clarified this in the revised manuscript (lines 116–117).*

Line 136: Why activity in this period is not considered as a seismic crisis? between 29-30 November you have even more events that in previous periods.

*A.: The term seismic crisis referred to the period with elevated seismicity beneath Brava, leading to evacuation of a village on Brava. The increased activity from 29 November to 2 December occurred offshore and the alert level for Brava was not raised.*

Line 226: In other cases (ex. El Hierro unrest) there was a seismic migration with several changes in path and direction with no systematic trend during some time periods.
See, C. López et al. (2017), Driving magma to the surface: The 2011–2012 El Hierro Volcanic Eruption, Geochem. Geophys. Geosyst., 18, 3165–3184, doi:10.1002/2017GC007023.

*A.: Thank you for pointing out this interesting reference. Although we think there is still some difference of the earthquake locations and shifts towards the place of eruption (El Hierro) and the place of the seismic crisis (Brava), we think that the interplay of tectonic and volcanic stresses could be a very interesting point. We extended the discussion accordingly (lines 246-251).*

**Changes made to figures:**

Figure 1: Bathymetry lines added
Figure 3: Accumulated number of events added
Figure 5: Bathymetry lines added
Figure 6: Figure added to include possible errors due to the outage of one array station
Figure 7 (formerly 6): Bathymetry lines added
Figure 8 (formerly 7): Addition of b–value estimation of the whole study period (a), and the swarms formerly shown in Fig. 10 (which we then deleted). Additionally, we added the number of events used for the estimation of b for a better understanding of the data availability.
Figure 9 (formerly 8): Bathymetry lines added
Figure 10 (formerly 9): Bathymetry lines added
Figure 11: Bathymetry lines added

[revised manuscript text omitted]